# Toxicological and Pharmacological Activities, and Potential Medical Applications, of Marine Algal Toxins

**DOI:** 10.3390/ijms25179194

**Published:** 2024-08-24

**Authors:** Xinyu Gao, Hanyi Wang, Kuilin Chen, Yifan Guo, Jin Zhou, Weidong Xie

**Affiliations:** 1State Key Laboratory of Chemical Oncogenomics, Shenzhen International Graduate School, Tsinghua University, Shenzhen 518055, China; gao-xy23@mails.tsinghua.edu.cn (X.G.); why19947680622@163.com (H.W.); ckl23@mails.tsinghua.edu.cn (K.C.); guo-yf23@mails.tsinghua.edu.cn (Y.G.); 2Shenzhen Key Laboratory of Health Science and Technology, Institute of Biopharmaceutical and Health Engineering, Shenzhen International Graduate School, Tsinghua University, Shenzhen 518055, China; 3Open FIESTA Center, Shenzhen International Graduate School, Tsinghua University, Shenzhen 518055, China; 4Institute for Ocean Engineering, Shenzhen International Graduate School, Tsinghua University, Shenzhen 518055, China

**Keywords:** marine algal toxins, bioactive components, toxicity, pharmacological activity, pharmacological mechanisms, environmental impact

## Abstract

Marine algal toxins have garnered significant attention in the research community for their unique biochemical properties and potential medical applications. These bioactive compounds, produced by microalgae, pose significant risks due to their high toxicity, yet offer promising therapeutic benefits. Despite extensive research identifying over 300 marine algal toxins, including azaspiracids, brevetoxins, cyclic imines, and yessotoxins, gaps remain in the understanding of their pharmacological potential. In this paper, we critically review the classification, bioactive components, toxicology, pharmacological activities, and mechanisms of these toxins, with a particular focus on their clinical applications. Our motivation stems from the increasing interest in marine algal toxins as candidates for drug development, driven by their high specificity and affinity for various biological receptors. We aim to bridge the gap between toxicological research and therapeutic application, offering insights into the advantages and limitations of these compounds in comparison to other bioactive substances. This review not only enhances the understanding of marine algal toxins’ complexity and diversity, but also highlights their extensive application potential in medicine and bioscience, providing a foundation for future research and development in this field.

## 1. Introduction

The ocean is one of the primary features of Earth’s surface, covering approximately 360 million square kilometers and accounting for about 71% of the planet’s surface area [1,2], not only providing a foundation for biodiversity, but also serving as a significant source of bioactive substances [3,4]. Among these, marine algal toxins, a variety of bioactive compounds produced by microalgae and accumulated in the marine food web [5], have attracted considerable attention in recent years due to their unique biochemical properties and potential medical applications. These toxins encompass a range of biologically active molecules, characterized by their distinct chemical properties and their high toxicity, both to various marine organisms and to humans [6].

Despite the growing recognition of marine algal toxins, a critical gap persists in understanding their pharmacological effects and potential therapeutic applications. While the toxicological impacts of these substances are well documented, there is an urgent need to explore their potential as novel therapeutic agents. This review addresses the fundamental question of why it is essential to study the pharmacological properties of marine algal toxins and their potential applications in medicine. Uncovering the therapeutic potential of these toxins could lead to significant advancements in drug development, providing new treatment strategies for various diseases.

The study of marine algal toxins began in the mid-20th century, when outbreaks of paralytic shellfish poisoning (PSP) were first linked to toxic algae [7,8]. The pioneering work of Yasumoto and colleagues in the 1980s identified saxitoxins as the causative agents of PSP [9]. Since then, researchers have identified over 300 different types of marine algal toxins, including saxitoxins, brevetoxins, and domoic acid. The discovery of these toxins has significantly advanced our understanding of marine toxicology and its impact on both human health and marine ecosystems [10]. Historical data show that the frequency and intensity of marine algal toxins have significantly increased over the past few decades. This trend has been observed globally, primarily due to rising ocean temperatures and increasing coastal eutrophication. Data from the United States from 1990 to 2019 indicate a notable rise in the annual occurrence of harmful algal bloom events in some coastal regions. Amnesic shellfish poisoning (ASP) events have frequently erupted along the West Coast, with emerging recurrent outbreaks also observed on the East Coast [11].

Previous reviews have extensively discussed various aspects of marine algal toxins, including their chemical structures, toxicological effects, and environmental impacts [12,13,14,15]. These reviews provide a solid basis for understanding the diversity of these toxins, but lack information on their pharmacological effects and potential medical benefits. For instance, while these studies effectively document the toxicological profiles of these compounds, they often fail to explore their therapeutic potential, a crucial aspect for drug development. Additionally, existing research primarily focuses on the adverse effects of these toxins, with insufficient emphasis on their beneficial bioactivities. This gap in the literature underscores the need for a comprehensive analysis that integrates toxicological findings with pharmacological applications. The novel contribution of this paper lies in its comprehensive integration of research results, focusing primarily on the potential pharmacological and therapeutic applications of marine algal toxins. This study seeks to bridge the gap between basic toxicological research and applied medicine by highlighting the potential of these toxins for new drug development and therapeutic strategies. By thoroughly examining the chemical components of these toxins and their interactions with human receptors, this paper aims to provide new insights into their bioactive properties and explore their potential as therapeutic agents.

The classification of marine algal toxins is mainly based on their chemical structures, sources, and bioactivities. Each toxin’s source, structure, and mechanism of action are different, and they have significant impacts on marine ecosystems and human health [16]. The chemical composition of marine algal toxins is diverse, mainly including cyclic peptides, ketones, aliphatic toxins, and alkaloids. The unique chemical structure of each toxin determines its distinct bioactivity and toxicity. By thoroughly studying these toxins’ chemical components, their toxic effects and potential pharmacological application values can be better understood.

In terms of pharmacological activities, due to the high specificity and affinity of marine algal toxins for various human receptors, they mainly act on the nervous and muscular systems, activating relevant important targets and regulating various physiological functions [17]. Algal toxins have been found to inhibit tumor cell growth, and even to induce apoptosis in cancer cells [18,19,20]. In various animal models and cell experiments, algal toxins have been confirmed to effectively reduce inflammatory responses, providing new treatment pathways for various inflammatory diseases, such as neurological and autoimmune diseases [21,22].

To date, more than 300 marine algal toxins and their derivatives have been identified, and the toxic effects produced by these distinct types of algal toxins are also different [23,24,25]. Clinical and toxicological research on marine algal toxins includes real clinical case applications, safety evaluations, toxin identification and metabolic mechanisms, dose–effect relationships, and toxicological evaluations [26]. In clinical scenarios, the applications of these toxins are mainly focused on drug development and treatment strategy exploration. Researchers are also studying the interactions between toxins and other drugs or disease states to ensure their safety in clinical applications. The identification of marine algal toxins and determination of their metabolic mechanisms are crucial for understanding their biological effects, which includes determining the specific chemical structures of toxins, metabolic pathways, and how they transform and eliminate in the human body. Dose–response relationship studies aim to determine how specific toxin doses affect organisms, including evaluating their potency and toxicity at different doses to establish safe usage ranges.

In this paper, we aim to review and discuss the classification, bioactive components, toxic effects and mechanisms, and pharmacological applications of marine algal toxins. Through in-depth analysis, we provide a comprehensive perspective on the complexity and diversity of marine algal toxins, revealing their broad application potential in the fields of medicine and bioscience, and promoting the comprehensive utilization and development of marine resources. The literature reviewed in this paper spans publications from 2000 to 2024, ensuring a comprehensive overview of the most relevant and recent findings in the field.

## 2. Chemical Classifications and Sources of Marine Algal Toxins

### 2.1. Main Components of Algal Toxins

Marine algal toxins have diverse chemical compositions. According to classification by the joint working group of the World Health Organization (WHO) and the Intergovernmental Oceanographic Commission of the United Nations Educational, Scientific and Cultural Organization (IOC-UNESCO) [27], the chemical components of marine algal toxins mainly include the following categories:

#### 2.1.1. Azaspiracids (AZAs)

Azaspiracids (AZAs) are polyether toxins with a unique spiro ring structure, primarily produced by the dinoflagellate genus Azadinium. They were first discovered in a food poisoning incident in Ireland in 1995 [28,29]. These toxins have carboxyl groups at their ends, making them acidic. Azaspiracids consist of 40 carbon atoms, with 20 stereoisomeric centers and nine ring structures [30]. Based on the functional groups at the C39 position of the AZA structure, azaspiracids can be divided into the 348 type (demethylated at the C39 position) and the 362 type (methylated at the C39 position). The main azaspiracids identified thus far include AZA-1, AZA-2, AZA-3, AZA-6, and AZA-7 [31,32]. AZA-1 has a molecular formula of C_47_H_71_NO_12_ and a molecular weight of approximately 841.5 Da; AZA-2 has a molecular formula of C_45_H_69_NO_12_ and a molecular weight of approximately 819.5 Da; AZA-3 has a molecular formula of C_43_H_67_NO_12_ and a molecular weight of approximately 797.5 Da. These compounds are highly toxic, inducing apoptosis and necrosis by affecting calcium ion flow and intracellular calcium ion levels.

#### 2.1.2. Brevetoxins (BTXs)

Brevetoxins (BTXs) are lipophilic cyclic polyether neurotoxins classified as neurotoxic shellfish poisons (NSPs), with about 70 derivatives identified to date. They are produced by dinoflagellates such as Karenia brevis [33]. The chemical structure of BTXs includes multiple conjugated double bonds and epoxy groups, and their structure comprises polyether and cyclic aldehyde groups, and their molecular weight ranges between 800 and 900 daltons. These toxins accumulate in shellfish, mainly in the forms of BTx-1 (Type A) and BTx-2 (Type B) [34,35]. BTx-1 (Type A) has a molecular formula of C_49_H_70_O_13_ and a molecular weight of approximately 894 Da, while BTx-2 (Type B) has a molecular formula of C_50_H_70_O_14_ and a molecular weight of approximately 910 Da. The mechanism of action of BTXs involves specific binding to the fifth site of voltage-gated Na^+^ channels, leading to persistent activation of the channel and allowing a large influx of Na^+^, causing depolarization of neuronal and muscle membranes and resulting in neurotoxic symptoms [36].

#### 2.1.3. Cyclic Imines (CIs)

Cyclic imines (CIs) are a recently discovered class of marine biotoxins characterized by their rapid toxicity, having caused characteristic rapid death in mice in intraperitoneal bioassays. They include various compounds produced by dinoflagellates, such as the genus Alexandrium. CIs are large cyclic compounds with an imine (carbon–nitrogen double bond) and spiro-ring ether groups, classified together because the imine group is their common pharmacophore and they demonstrate similar intraperitoneal toxicity in mice [37,38]. The molecular weight of CIs ranges between 300 and 500 daltons, mainly including spirolides and gymnodimines. Spirolide B has a molecular weight of 411.6 Da and contains a cyclic amine group and multiple spiro-ring ethers in its structure. Gymnodimine A has a molecular formula of C_32_H_45_NO_4_ and a molecular weight of approximately 507.7 Da. CIs are known for their rapid-onset neurotoxic effects, blocking acetylcholine receptors and affecting neural signal transmission.

#### 2.1.4. Domoic Acids (DAs)

Domoic acids (DAs), mainly produced by diatoms such as the genus Pseudo-nitzschia, are water-soluble and have strong neurotoxicity; their chemical structure is similar to that of glutamic acid, and they can mimic glutamic acid in the central nervous system, leading to overexcitation and damage to neurons [39]. DAs have a molecular weight of approximately 311.1 Da, containing a cyclohexane ring and multiple carboxyl groups. Domoic acid is the representative compound of this class of toxins, with a molecular formula of C_15_H_21_NO_6_. After exposure, domoic acid primarily binds to N-methyl-D-aspartate (NMDA) receptors in the central nervous system, causing neuronal depolarization and leading to amnesic shellfish poisoning (ASP) [40].

#### 2.1.5. Okadaic Acids (OAs)

Okadaic acids (OAs), produced by dinoflagellates such as the genus Dinophysis, are lipid soluble, water insoluble, and heat stable. Their chemical structure is a complex polyketide system, primarily causing cytotoxicity by inhibiting protein phosphatases (PP1 and PP2A) [41]. OAs have a molecular formula of C_44_H_68_O_13_ and a molecular weight of 805.1 Da, containing multiple cyclic ester and carboxyl groups. Okadaic acid is the main representative of this class of toxins, causing cytoskeletal disruption and diarrhetic shellfish poisoning (DSP) by inhibiting protein phosphatases. Recent studies suggest that the toxic effects of OAs may be related to alterations in the absorption of nutrients, ions, and water in the small intestine through changes in the transport system [42].

#### 2.1.6. Pectenotoxins (PTXs)

Pectenotoxins (PTXs) are produced by dinoflagellates such as the genus Dinophysis. PTX-1 and PTX-2 were first isolated from the hepatopancreas of the Japanese scallop Patinopecten yessoensis by Japanese scientists Yasumoto et al. in the 1980s. Fourteen PTX congeners have been identified, including PTX-1, -2, -3, -4, -6, -7, -8, -9, -11, -12, -13, and -14, while PTX-5 and PTX-10 have not yet been reported [43]. The chemical structure of PTXs is complex, containing multiple cyclic ethers and ester bonds, with molecular weights ranging between 800 and 900 Da. PTX-2 has a molecular formula of C_47_H_70_O_14_ and a molecular weight of 876.1 Da, with its structure containing multiple cyclic ester and lactone groups. PTXs primarily cause cytotoxicity by affecting the integrity of the cytoskeleton and cell membrane.

#### 2.1.7. Saxitoxins (STXs)

Saxitoxins (STXs) are synthesized primarily by toxic dinoflagellates, including genera such as Gymnodinium, Alexandrium, and some freshwater algae, and accumulate in shellfish through the food chain [44]. Saxitoxin (STX) is one of the most toxic known marine biotoxins. As the earliest identified primary toxin responsible for paralytic shellfish poisoning (PSP), it was first identified in the highest concentrations in large stonefish from Alaska, USA [45]. Its chemical structure contains multiple guanidine groups, making it a potent neurotoxin. The chemical structure of STX was formally identified in 1975, with a molecular formula of C_10_H_17_N_7_O_4_ and a molecular weight of 299. It belongs to the family of marine guanidine toxins, being a tetrahydropurine derivative with active sites primarily at two guanidine groups and two hydroxyl groups [46].

#### 2.1.8. Yessotoxins (YTXs)

Yessotoxins (YTXs), first discovered in Japan in 1986, are produced by the dinoflagellate Protoceratium reticulatum of the family Gonyaulacaceae [47,48]. The chemical structure of YTXs is complex, containing multiple cyclic ethers and ester bonds. Research has shown that the toxic components produced by several marine dinoflagellates are primarily YTXs, consisting of 11 contiguous ether rings forming a ladder structure [49]. YTXs have a molecular formula of C_55_H_82_O_21_S_2_Na_2_ and a molecular weight of approximately 1141.36 Da. They cause cytotoxicity and cardiovascular systemic effects by interfering with intracellular calcium ion balance and the cytoskeleton. The classifications and introductions of these marine algal toxins are shown in Table 1.

### 2.2. Toxin Production and Influencing Factors

Understanding the mechanisms and factors behind the production of algal toxins can help formulate effective prevention and control strategies to reduce the harm of algal toxins to the ecological environment and human health.

The production of marine algal toxins is influenced by various factors, mainly including the type of algae, environmental conditions, and ecological interactions. Different types of algae produce different types of toxins. Cyanobacteria mainly produce cyclic peptide toxins (e.g., microcystins) and alkaloid toxins (e.g., cylindrospermopsins). Dinoflagellates produce a variety of toxins, including azaspiracids (AZAs), brevetoxins (BTXs), and yessotoxins (YTXs). Diatoms are primarily associated with carbamate toxins (e.g., domoic acids). Algae produce toxins through complex biosynthetic pathways, and the types and concentrations of toxins vary depending on the species of algae.

The generation of algal toxins is regulated by various environmental factors, including temperature, light, and nutrient concentrations [1]. Most algae grow rapidly and can produce more toxins at higher temperatures; for instance, warm sea temperatures often promote the proliferation of dinoflagellates, thereby increasing the concentration of algal toxins. Photosynthesis is the basis for algal growth, so light intensity has a significant impact on the production of algal toxins; higher light intensity generally promotes algal growth and toxin accumulation [50]. LED light sources, with light intensity of 200 μmol photons m^2^/s, were used in a photocycle of 12 h light/12 h dark, as such light conditions are widely believed to promote photosynthesis in algae and directly affect toxin production [6], thus, an increase in the intensity of light can lead to an increase in the concentration of algal algins. The concentrations of nutrients, such as nitrogen and phosphorus in the water, directly affect algal reproduction and toxin production. In eutrophic waters, abundant nutrients often lead to massive algal proliferation and toxin accumulation. Still or slow-flowing water is conducive to algal aggregation and toxin production, while strong water flow may disperse algae and reduce toxin concentrations. Pollution sources such as industrial wastewater, agricultural runoff, and domestic sewage also increase the nutrient levels in water bodies, promoting excessive algal growth and frequent algal toxin pollution events [6,51]. In BG11 standard medium with concentrations of 2.5 mg/L for nitrogen and 0.5 mg/L for phosphorus, the concentrations of nutrients directly affected the growth rate of algae and the production of toxins. Such experiments have shown that excess nitrogen can promote the rapid reproduction of certain algae and increase their toxin production [11].

Ecological interactions between algae and other organisms also affect toxin production. Algae may increase toxin production in response to predators in order to enhance their defense mechanisms and reduce the risk of predation. Competition among algae can also influence toxin production; some algae release toxins to inhibit the growth of competitors, thus gaining an advantage in the habitat. On the other hand, certain algae have symbiotic relationships with other micro-organisms or organisms, which may promote or inhibit toxin production, for example, some bacteria symbiotic with algae provide nutrients or growth hormones, promoting algal growth and toxin accumulation [52].

In this section, the chemical classification and origin of marine algal toxins are discussed in detail. The chemical compositions of marine algal toxins are very diverse, involving multiple classes of toxins, such as polyether toxins, cycloaminotoxins, carbohydrate toxins, etc. These toxins are mainly produced by different types of algae (such as dinoflagellates, cyanobacteria, and diatoms), with different chemical structures and toxic mechanisms. Through in-depth analysis of the major toxin classes, we not only reveal the complex chemical properties of these toxins, but also explore their sources in the ecosystem and their potential threats to human health. Further understanding of the classifications and sources of these toxins will help in developing more effective prevention and control strategies in order to reduce the harm of algal toxins to the ecological environment and public health.

## 3. Toxicity and Mechanisms of Marine Algal Toxins

After describing in detail the chemical classification of marine algal toxins and the specific sources of each toxin, we will further explore how these toxins affect biological systems. The unique chemical properties of each toxin not only determine their interaction with living organisms, but also affect the distribution and stability of toxins in the environment, which in turn affects their accumulation in the food chain. The following details how these toxins affect the nervous system, liver, gastrointestinal tract, and immune system through specific biochemical pathways, demonstrating their toxicological effects and potential health risks.

### 3.1. Neurotoxicity

Marine algal toxins exhibit significant neurotoxicity, including neuroparalysis, ataxia, memory loss, neuralgia (pain), and neurodegenerative diseases. Recent studies have provided deeper insights into the mechanisms of action of these toxins. It has been shown that marine algal toxins often target ion channels, neurotransmitter receptors, and signaling pathways. They can disrupt ion homeostasis, induce oxidative stress, and trigger apoptotic and necrotic cell death [53,54]. Many of these toxins activate voltage-gated sodium channels or interfere with neurotransmitter receptors, such as nicotinic acetylcholine receptors, leading to altered neuronal excitability and neurotransmission [55]. Furthermore, marine algal toxins may also modulate the expression of genes involved in neuroinflammation, exacerbating neural damage through inflammatory responses. These findings highlight the complexity of the neurotoxic effects of marine algal toxins and underscore the need for continued investigation into their precise mechanisms of action [56,57]. The neurotoxic effects of different marine algal toxins produced by various algae are detailed below:

Azaspiracids (AZAs), often contaminating seafood, demonstrate multi-system toxicity, particularly severe neurotoxicity. AZA-1 can increase lactate dehydrogenase release in mouse cortical neurons, induce nuclear condensation, and stimulate caspase-3 activity, indicating that it can cause cortical neuron death by inducing both apoptosis and necrosis [58]. Oral administration of moderate doses of AZA-1 (100–300 μg/kg) generally does not cause significant poisoning symptoms, but high doses (420–780 μg/kg) can lead to dose- and time-dependent poisoning symptoms in mice, such as lethargy and reduced movement [59]. The final symptoms of mice orally administered with AZAs include inactivity, lateral recumbency, tremors, abdominal breathing, hypothermia, and cyanosis [60].

Brevetoxins, primarily produced by the dinoflagellate Karenia brevis, are lipophilic cyclic polyether neurotoxins [34]. The neurotoxicity of brevetoxins includes ataxia, but generally does not include paralysis. Fish and other animals consuming these algae can become poisoned, and the toxins can accumulate up the food chain, eventually affecting humans. Consumption of shellfish contaminated with brevetoxins can cause neurotoxic shellfish poisoning (NSP) in humans within 30 min to 3 h, presenting as a series of central and peripheral nervous system-related toxic reactions [61]. Brevetoxins have a high affinity for the fifth site of voltage-sensitive sodium (Na^+^) channels, activating the channels and increasing membrane permeability to sodium ions, subsequently leading to membrane depolarization and altered neuromuscular transmission [62].

Cyclic imines primarily block acetylcholine receptors (both nicotinic and muscarinic) in the central and peripheral nervous systems, thereby hindering neural transmission. The antagonistic effect on nicotinic acetylcholine receptors (nAChRs) disrupts neuromuscular transmission, causing symptoms such as paralysis and respiratory difficulty. In mice, cyclic imines show extremely high acute toxicity via intraperitoneal injection, causing either rapid death within 20 min or full recovery without noticeable aftereffects [63].

Paralytic shellfish toxins (PSPs) are potent neurotoxins that cause paralysis by blocking voltage-gated sodium channels [64,65]. One key toxin, saxitoxin, exhibits extreme neurotoxicity and is produced by several freshwater cyanobacteria and marine dinoflagellates [66,67]. Saxitoxin blocks voltage-gated sodium channels, preventing normal cell function and causing paralysis [7,10].Tetrodotoxin (TTX), another potent neurotoxin, is about 1250 times more toxic than cyanide and specifically binds to voltage-gated sodium channels. While often associated with pufferfish, it is not produced by the pufferfish themselves, but is acquired through the food chain. Certain bacteria, such as those in the genera Vibrio and Pseudomonas, produce TTX [68]. Additionally, some algae, including certain cyanobacteria and red algae, can produce TTX [69]. The toxicological effects of saxitoxin are similar to those of TTX, inhibiting neural transmission and causing paralysis and, potentially, death [2,67]. Saxitoxin’s high toxicity is evidenced by an LD50 of 10 μg/kg in mice, though it can also provide potent local anesthetic and analgesic effects at lower doses, offering potential value for new drug development [70].

Domoic acid, a water-soluble neurotoxin mainly produced by diatoms of the genus Pseudo-nitzschia and red algae of the genus Chondria, is the main component of amnesic shellfish poisoning (ASP), which can damage memory regions like the hippocampus [71]. Intraperitoneal injection (IP) of DA at doses of 1.0–2.5 mg/kg bw in adult and elderly C57Bl/6 NIA mouse models highlighted that high doses (2.0–2.5 mg/kg bw) caused severe tonoclonus (CTC) in elderly female mice that eventually required premature euthanasia [72]. Studies have shown that domoic acid significantly causes neuron loss in the hippocampus of mice, with reduced expression of proteins related to learning and memory [73]. The main toxin contaminating shellfish and crustaceans is domoic acid, with the highest toxicity among its isomers, and an LD50 of about 2.5 mg/kg in rats [3]. The neurotoxic mechanism of domoic acid involves excitatory amino acid receptors and synaptic transmission [71]. Once ingested, domoic acid distributes as charged hydrophilic molecules in vascular tissues, leading to neurotoxicity. It significantly decreases mitochondrial oxidative respiratory chain function in mouse hippocampal neurons, increasing ATP consumption and reducing the expression of PGC-1α, thereby inhibiting mitochondrial proliferation pathways and causing mitochondrial dysfunction. Domoic acid significantly increases ROS and protein carbonylation levels in mouse hippocampal tissue, causing oxidative stress [74]. It also upregulates NADPH oxidase expression, further damaging nervous tissue. Pathological examination of individuals who died from domoic acid poisoning showed hippocampal and thalamic damage leading to memory loss, hence the term “amnesic shellfish poisoning” [75].

Ciguatoxins, produced by benthic microalgae, are potent neurotoxins known to be among the most toxic to mammals, with an LD50 of only 0.05 μg/kg—higher than other dinoflagellate toxins like brevetoxins and tetrodotoxins [5,76]. Symptoms of ciguatoxin poisoning include limb tingling or cold allodynia, with severe cases resulting in cardiovascular and respiratory failure leading to coma and death. These toxins have high affinity for voltage-gated Na^+^ channels, can cross the blood–brain barrier, and affect the central nervous system [77].

Marine cyanobacteria are known to produce a variety of structurally diverse toxins, including neurotoxic substances [78], that are associated with both acute and chronic health impacts, with long-term exposure linked to neurodegenerative diseases such as Alzheimer’s and Parkinson’s. The health effects of common cyanobacterial toxins range from mild rashes and gastrointestinal discomfort to severe liver damage and neurotoxicity; in extreme cases, they can be fatal [78].

Palytoxin (PLTX) induces cell death and related physiological reactions by activating Ca^2+^-permeable non-selective cation channels, causing a rapid increase in intracellular Ca^2+^ concentrations. PLTX also activates calcium-activated proteases in the cytosol, leading to cell necrosis [79,80]. In mice, the LD50 of PLTX is only 50 mg/kg, and an intraperitoneal injection of just 0.13 μg/kg can be lethal [81].

While significant progress has been made in understanding the neurotoxic effects of marine algal toxins, much remains to be explored. The detailed mechanistic studies provide a foundation for future research aimed at mitigating the health risks associated with these toxins. Additionally, the potential for these toxins to contribute to chronic neurological conditions warrants further investigation, particularly in light of increasing environmental exposure risks. Table 2 summarizes each marine algal toxin, its source, and its primary neurotoxic effects.

### 3.2. Hepatotoxicity

Microcystins are cyclic heptapeptide toxins mainly produced as natural metabolites by cyanobacteria (e.g., Microcystis, Anabaena) in aquatic ecosystems. Humans can be exposed to microcystins by consuming contaminated water, fish and shellfish, vegetables, or algae-based dietary supplements, as well as through recreational activities. Microcystin-LR (MCLR) is a typical microcystin, reported to be the most common and toxic variant, and is the only microcystin with a defined daily tolerable intake (0.04 μg/kg) [82]. Microcystins cause hepatotoxicity by inhibiting protein phosphatases, specifically serine/threonine phosphatases, thereby leading to disruptions in critical cellular processes and resulting in hepatocyte damage [83]. This damage can trigger acute or chronic liver diseases, including hepatitis and cirrhosis. The hepatotoxicity of microcystins reflects their severe impacts on liver function, with microcystin-LR being the most extensively studied among them; its inhibition of protein phosphatases A1 and A2 disrupts intracellular signaling, causing cell death and tissue damage [84,85]. These toxins can lead to direct hepatocyte damage and liver inflammation, with long-term exposure potentially causing liver dysfunction, cirrhosis, or liver cancer.

### 3.3. Gastrointestinal Toxicity

Some algal toxins, when ingested through the food chain, can cause gastrointestinal symptoms such as nausea, vomiting, and diarrhea [86]. These symptoms highlight the direct impacts of algal toxins on the digestive system, primarily caused by diarrhetic shellfish poisoning.

Azaspiracids (AZAs) are marine algal toxins that accumulate in edible shellfish, primarily affecting the gastrointestinal tract and causing foodborne gastrointestinal poisoning in humans. The toxicity of different AZA toxins varies; the oral acute lethal doses (LD50s) of AZA-1, AZA-2, and AZA-3 in mice are 443 μg/kg, 626 μg/kg, and 875 μg/kg, respectively. The minimal lethal doses (MLDs) for intraperitoneal injection in mice are 150 μg/kg, 110 μg/kg, and 140 μg/kg, respectively [87]. LC-MS/MS analysis of major organs shows that the absorption of these toxins through the gastrointestinal tract is dose-dependent [60]. After AZA-1 poisoning, the recovery of mice is very slow, with gastric and intestinal erosion, and shortened villi, persisting for up to 3 months, and alveolar wall edema and hemorrhage symptoms can last for 56 days [88].

Cyclic Imines (CIs), produced by some dinoflagellates, are marine biotoxins known for their “rapid action toxicity”, causing rapid death in animal models, like in mice via intraperitoneal injection. Pinnatoxins (PnTXs), isolated from shellfish, exhibit high toxicity, with LD99s of 180 μg/kg in Japanese shellfish and 135 μg/kg in New Zealand green-lipped mussels and oysters for PnTXA, as well as 22 μg/kg for PnTXB/C and 400 μg/kg for PnTXD in Japanese shellfish [37].

Okadaic acid (OA), an effective inhibitor of protein phosphatases 1 and 2A, causes diarrhea and gastrointestinal symptoms when ingested through contaminated shellfish. Studies show that OA stimulates cell movement and causes loss of focal adhesion stability and cytoskeletal disorganization due to altered tyrosine phosphorylation states for focal adhesion kinase and paxillin. OA causes cells to round up and lose barrier properties through mechanisms possibly involving F-actin disruption and/or over-phosphorylation and activation of kinases that stimulate tight junction breakdown [89]. There are no reported fatal cases in humans, but long-term accumulation may have carcinogenic effects. Studies show that the acute oral LD50 of OA and its derivative, DTX-1, are 1069 μg/kg and 897 μg/kg, respectively [90,91]. Moreover, acute toxicity in mice indicates that esterified toxins like OA have higher toxicity than free toxins [92].

Pectenotoxins (PTXs), produced by dinoflagellates (e.g., Dinophysis), accumulate in shellfish through the food chain, affecting humans and other organisms that consume these contaminated shellfish. PTXs have been found to exhibit significant toxicity in various cells in mice, particularly causing gastrointestinal toxicity [93]. In mouse toxicity tests, intraperitoneal injection of different PTX varieties showed varying minimal lethal doses (LD50s), ranging from 1924 μg/kg to 770 μg/kg mouse weight for PTX-1, PTX-2, PTX-3, PTX-4, PTX-7, and PTX-11, and higher than 5000 μg/kg for PTX-8, PTX-9, and PTX-2 SA. Studies indicate that oral doses of 1500 μg/kg PTX-2 did not cause diarrhea in mice, but led to small intestine fluid accumulation, while doses of 2000 μg/kg caused diarrhea [94]. However, oral doses of 5000 μg/kg PTX-2 or PTX-11 did not cause diarrhea. In vitro toxicity tests showed no toxicity of PTX-2 SA at 1.8 μg/L to KB cells, while PTX-2 exhibited significant toxicity at 0.05 μg/L [95].

Yessotoxins (YTXs) are also typical diarrhetic shellfish toxins that damage cells through multiple mechanisms, including programmed cell apoptosis, non-programmed cell death, or autophagy, ultimately contributing to YTX gastrointestinal toxicity. The sensitivity of YTX toxic effects varies among cell types, indicating different cellular responses to YTX [96,97]. YTX toxicity is mainly associated with dysregulation of intracellular calcium ion balance. YTX promotes the opening of mitochondrial permeability transition pores, leading to the release of apoptotic factors into the cytoplasm, triggering cell death. It also facilitates the influx of extracellular calcium ions through voltage-gated calcium channels, disrupting intracellular calcium ion homeostasis and impairing normal cellular signaling [98]. This intracellular calcium ion dysregulation further reduces cell adhesion, weakening intercellular adhesion and affecting cell polarity and structural integrity. YTX also regulates intracellular cyclic adenosine monophosphate (cAMP) levels, influencing cAMP-related kinase family proteins involved in various cellular functions, including proliferation, death, and mobility. Finally, YTX causes depolymerization of cytoskeletal proteins, such as F-actin filaments and tensin proteins, further disrupting cellular structure and function [99,100,101,102]. These mechanisms directly impact gastrointestinal cell survival and function, leading to gastrointestinal toxicity, including symptoms like abdominal pain, nausea, vomiting, and diarrhea. Current research shows a wide range of acute intraperitoneal LD50 values for YTX, from 80 to 750 μg/kg·bw [103].

Toxins from harmful algal blooms (HABs) can enter the human body through the food chain [104,105]. Cylindrospermopsin, produced by several freshwater cyanobacteria, damages intestinal epithelial cells, interfering with normal digestive processes and causing gastrointestinal reactions [106]. Microcystins (MCs), secondary metabolites produced by cyanobacteria, vary in levels across different sources of drinking water, and their exposure levels are positively correlated with colorectal cancer incidence [107].

### 3.4. Dermatotoxicity

Contact with seawater or seafood containing algal toxins can lead to skin inflammation and allergic reactions, presenting as rashes, itching, and edema [86,108]. These symptoms reveal that algal toxins can cause direct skin irritation and allergic reactions, especially when individuals are exposed to water containing microcystins.

Microcystins (MCs) and cylindrospermopsin (CYN) have been reported to cause these symptoms through skin contact. Microcystins disrupt cell membrane integrity, leading to cell damage and inflammatory responses, thereby causing skin irritation and allergic reactions [109]. Studies have shown that microcystins significantly reduce collagen fibrils in the dermal layer of skin tissue, downregulate the transcription of genes related to tight junctions and the stratum corneum, and damage the physical barrier of the skin. Tadpoles exposed to microcystins exhibit increased eosinophils and upregulated transcription of inflammation-related genes, emphasizing that exposure to environmental concentrations of microcystins can lead to skin inflammation [110]. Aplysiatoxin (ATX) derivatives, a class of polyketide structural skin toxins with ion channel inhibitory effects, exhibit their dermatotoxicity by inhibiting the Kv1.5 ion channel subtype [111,112]. Studies have found that ATX derivatives containing the PKC kinase recognition region can phosphorylate PKCδ kinase and inhibit Kv1.5 ion channel currents, such as debromoaplysiatoxin and neo-debromoaplysiatoxin A [113]. Despite lacking PKC kinase phosphorylation activity, oscillatoxin derivatives with only AB mono-oxygen spiro rings still significantly inhibit Kv1.5 channel currents, such as 30-methyloscillatoxin D and neo-debromoaplysiatoxin B, indicating that ATX derivatives may inhibit Kv1.5 potassium ion channel currents by activating PKC kinase or directly blocking ion channels, thereby causing dermatotoxic reactions [114]. The mechanisms of action of ATX derivatives, including ion channel current inhibition, reveal their role in skin toxicity. Studies have shown that CYN and its degradation products significantly inhibit keratinocyte proliferation at concentrations corresponding to naturally occurring levels of CYN (1 μg/mL), with pure CYN demonstrating 70% inhibition within 24 h. The cytotoxic effects of CYN and its degradation products on keratinocytes are also significant, with the pure toxin (1 μg/mL) showing an estimated 35% inhibition after 24 h of exposure. Harmful effects of CYN and its byproducts were also observed during keratinocyte migration, with the initial form of the toxin (1 μg/mL) showing 40% inhibition within 16 h [115].

### 3.5. Immunotoxicity

Some algal toxins affect the human immune system by inhibiting normal function and reducing resistance [116]. This inhibition of immune system function can make individuals more susceptible to other diseases. For example, microcystins can affect lymphocyte activity and suppress immune responses, revealing that algal toxins can interfere with the normal function of immune cells, thereby reducing the body’s resistance to pathogens and increasing susceptibility to other diseases [110]. Cylindrospermopsin (CYN) disrupts immune cell function by inducing oxidative stress and DNA damage [117]. CYN significantly affects the pro-inflammatory mediator tumor necrosis factor-alpha (TNF-α), enhancing the effects of bacterial and cyanobacterial LPS. It activates inflammatory signaling pathways, including p38 mitogen-activated protein kinase, and subsequently induces the expression of inducible nitric oxide synthase (iNOS) and the production of pro-inflammatory mediators, such as nitric oxide (NO), TNF-α, and interleukin-6 (IL-6), demonstrating its impact on immune cell activity [118]. These mechanisms collectively lead to decreased immune system function, further increasing disease susceptibility.

In summary, algal toxins exhibit extensive toxicity to multiple organs, including the nervous system, gastrointestinal tract, liver, immune system, and skin. These toxic effects can be caused by various types of algal toxins, and some algal toxins exhibit multiple toxic effects mediated by different mechanisms (Figure 1).

This section explores the broad toxic effects of marine algal toxins on a variety of biological systems in detail, including the nervous system, liver, gastrointestinal tract, skin, and immune system. These toxins cause severe neurotoxicity, hepatotoxicity, gastrointestinal toxicity, dermatotoxicity, and immunotoxicity through a variety of mechanisms, such as interfering with ion channels, inducing oxidative stress, and apoptosis. Different types of algal toxins exhibit unique toxicity mechanisms. Algal toxins can also exacerbate tissue damage and disease progression by regulating gene expression and inflammatory responses. By summarizing these toxicological mechanisms, we recognize the potential threat of marine algal toxins to ecosystems and human health, highlighting the importance of in-depth research into the mechanisms of action for these toxins. Future studies should continue to explore the molecular mechanisms of these toxins in order to develop effective protective measures and therapeutic strategies.

## 4. Pharmacological Activity and Mechanisms of Marine Algal Toxins

Although Marine algal toxins exhibit significant toxicity in biological systems, the complex biochemical properties of these toxins also provide potential opportunities for their application in the field of pharmacology. After exploring how these toxins affect the nervous system, liver, gastrointestinal tract, and immune system through specific biochemical pathways, we will further investigate their pharmacological potential. Some algal toxins have been found to have a variety of potential medical benefits, such as anti-tumor, anti-inflammatory, and so on, which highlights the possibility of using these toxins as candidate molecules for new drug development. Subsequent sections will detail the pharmacological activity of marine algal toxins and their effects on specific types of cancer cells, demonstrating their potential applications in modern medicine, in addition to being toxic substances.

### 4.1. Anti-Tumor Activity

Paralytic shellfish toxins (PSPs) are a class of neurotoxins that have shown anti-cancer effects. In a study on naturally occurring canine hemangiosarcoma, high doses of PSPs significantly delayed the progression of abdominal metastasis. In all dosage groups, PSPs did not cause any unexpected complications, and the treated dogs showed a longer survival period compared to the longest median survival time reported in the literature [18]. Another study investigated the effects of PSPs on the function of monocytes in human peripheral blood mononuclear cells (PBMCs) and found that PSPs significantly increased the number of CD14(+)/CD16(−) monocytes at a concentration of 100 μg/mL, without significantly affecting the proliferation activities of T cells, NK cells, or B cells across a concentration gradient of 1–100 μg/mL [19].

Okadaic acid (OA) is an algal toxin widely present in the marine environment, primarily produced by certain species of planktonic algae. OA induces tumor cell apoptosis by inhibiting protein phosphatases 1 (PP1) and 2A (PP2A), thereby interfering with cell cycle regulation. This mechanism suggests that OA engages in potential anti-cancer activity against certain types of cancer. The half-maximal effective concentration (EC50) of OA on Caco-2 cells is 48.8 nM, however, the apparent permeability coefficient (Papp) of OA in Caco-2 cell monolayers is less than 1.0 × 10^−6^ cm/s, indicating low permeability and that, although OA has low permeability in the intestinal environment, its high toxicity may still exert effects on cancer cells in localized environments [20]. Moreover, OA and its analogs, Dinophysistoxin-1 (DTX-1) and Dinophysistoxin-2 (DTX-2), have been studied for their abilities to induce apoptosis in tumor cells through the same inhibitory effects on PP1 and PP2A. DTX-1 demonstrated even greater cytotoxicity than OA, with EC50 values of 5.46 nM in Neuro-2a neuroblastoma cells, 5.66 nM in NG108-15 hybridoma cells, and 12.27 nM in MCF-7 breast cancer cells. In contrast, DTX-2 exhibited lower cytotoxicity, with EC50 values of approximately half those of OA in Neuro-2a and NG108-15 cells, and 64.71 nM in MCF-7 cells [119].

Botana and colleagues compared the apoptosis-inducing effects of YTX on 58 human tumor cell lines and found that some tumor cell lines were particularly sensitive to YTX. For example, melanoma cells, lung cancer cells, colon cancer cells, and acute leukemia cells showed high sensitivity to YTX. In contrast, breast cancer cell lines had moderate sensitivity to YTX, while ovarian, kidney, central nervous system, and prostate tumor cell lines exhibited lower sensitivity. Additionally, non-tumor cell models showed significantly lower sensitivity to YTX toxicity compared to tumor cells [21]. In an in vivo study, Araceli et al. tested the anti-tumor effects of YTX using a B16F10 murine melanoma model. They injected a high dose, 100 μg/Kg YTX, near the tumor on the first day, followed by lower doses of YTX from the second to the fifth day. The results showed significant improvement around the established tumor tissue without displaying toxicity. Based on these results, YTX is considered a potential new tool for cancer treatment [22]. Table 3 provides a summary of the pharmacological activities and mechanisms of various marine algal toxins against different cancer cell lines.

### 4.2. Analgesic Effects

Tetrodotoxin (TTX) is recognized as the primary toxin in pufferfish, and recent studies have shown that it can also be produced by dinoflagellates [120]. TTX, as a potent neurotoxin, has exhibited significant analgesic effects in various neuropathic pain models [68]; it specifically binds to voltage-gated sodium channels (VGSCs), physically blocking the flow of sodium ions through these channels, thereby preventing the generation and propagation of action potentials (APs). In conditions of pathological pain, such as neuropathic pain, there is a notable re-expression of embryonic VGSC subtype NaV1.3 in adult primary sensory neurons; the upregulation of these TTX-sensitive VGSCs leads to hyperalgesia, while TTX achieves analgesia by blocking VGSCs [121]. Nieto et al. conducted clinical research on the effects of TTX on pain symptoms, showing that TTX administration at doses of 15–90 µg/day via intramuscular injection significantly reduced patients’ pain levels, with only transient and mild toxicity. An optimal analgesic regimen was found to be intramuscular injection of 30 µg TTX twice daily for four consecutive days. Additionally, the presence of TTX-based painkillers/anesthetics on the market further supports the clinical application of TTX in pain management [122]. Recent studies have further highlighted the potential of TTX in managing cancer-related pain, showing that intramuscular injections of TTX at doses of 30 µg twice daily significantly reduced pain in cancer patients with neuropathic pain, with pain scores decreasing by more than 50% in the majority of patients (*n* = 40) in clinical trials [123]. The combination of TTX with other analgesics, such as opioids, has demonstrated enhanced pain relief, allowing for lower doses of opioids and reducing their associated side effects [122]. Research also suggests that TTX might have neuroprotective properties. In a mouse model of cerebral ischemia/reperfusion injury, TTX administration at a dose of 10 µg/kg significantly reduced infarct size by 30% and decreased markers of oxidative stress and inflammation [124], making TTX a promising candidate for treating various neurological disorders, including stroke and neurodegenerative diseases, however, there are some potential adverse effects of using TTX in a therapeutic setting. TTX is a highly toxic alkaloid neurotoxin that can have a fatal effect on the human body when ingested in excess of a lethal dose. Clinical trials have shown that, when TTX is used to relieve cancer-related pain, some patients may experience mild-to-moderate transient adverse effects, such as nausea, dizziness, numbness, or tingling in the mouth [125]. Therefore, the use of TTX requires strict control of dose and route of administration to ensure its safety and efficacy. Future studies are needed to optimize the therapeutic formulation and administration strategy of TTX, reduce its potential risk of toxicity, and ensure its safety in clinical use.

Saxitoxin (STX) is a paralytic shellfish toxin that belongs to the alkaloid class of neurotoxins [126]. In one study, zebrafish were administered saxitoxin at a dose of 7.61 μg eq/kg body weight via intraperitoneal injection, with acetic acid solution as a control. Testing was conducted at various time points within 24 h post-treatment. The results showed that saxitoxin could inhibit the transmission of pain signals by reducing glutamate (Glu) levels, while also exerting analgesic effects through the upregulation of P and β-endorphin (β-EP) [127].

STX has also shown significant analgesic effects in mice. In one experiment, mice were administered an intraperitoneal injection of 200 μL of saxitoxin, equivalent to 5.12 μg STX per kg body weight. Thirty minutes later, the mice were given another intraperitoneal injection of 200 μL of 0.6% acetic acid solution to calculate the inhibition rate of the writhing response caused by saxitoxin. The mice were then placed on a hot plate apparatus, and the time it took for them to exhibit paw-licking or jumping responses (pain threshold) was recorded. The results showed that the control group had an average of 56.60 ± 2.50 writhing episodes, while the experimental group had only 18.20 ± 1.83 episodes—a significant reduction in the experimental group compared to the control group (*p* < 0.001). The inhibition rate of the writhing response after saxitoxin administration was 67.84%. In the hot plate test, the pain threshold of the control group was 20.60 ± 1.66 s, whereas the experimental group had a pain threshold of 35.53 ± 0.48 s, significantly higher than the control group (*p* < 0.001). After saxitoxin administration, the pain threshold increased by 72.48% [128].

### 4.3. Immunomodulatory Effects

Alicia et al. explored the immunomodulatory potential of marine algal yessotoxin (YTX) on the mouse T lymphocyte line EL-4. The results indicated that YTX might influence TCR expression levels through the PKC and PP2A pathways (with 100 nM YTX showing the most significant effect on TCR expression), thereby exhibiting certain regulatory abilities on T cell activity. Additionally, existing research has demonstrated that YTX is effective at treating allergies and allergic asthma. Patent EP1875907 discloses a method of using YTX to treat allergies and asthma by regulating the immune effects of mast cells in mice to inhibit histamine release, thus controlling allergic reactions [129].

Brevetoxin (PbTX) can modulate the immune response in humans and animals; its mechanism involves activating immune cells, stimulating the production of γ-globulin, cytokines, and neutrophils, regulating lysozyme activity, inducing apoptosis, and modulating lymphocyte proliferation in marine species. In a study by Brammer-Robbins et al., PbTx-2 and PbTx-3 in low concentration ranges (PbTx-2: 10^−5^ to 10^−12^ M, PbTx-3: 10^−4^ to 10^−12^ M) were found to produce certain immunomodulatory effects on the Jurkat human leukemia T cell line. This immunomodulatory effect mainly manifests at low concentrations, thus exhibiting a lower toxicity while still being able to change the physiological state of cells by affecting cell metabolism and regulating the activity of immune cells. Low concentrations of PbTx-3 induce different immune responses, such as regulating cytokine and lymphocyte proliferative activity [130].

In addition to the extensively studied algal toxins mentioned above, there are still many algal toxins with relatively unexplored pharmacological applications. To date, the relationships between the pharmacological activities of algal toxins and their safety are still not fully understood. The structural differences among various algal toxins necessitate specific safety testing systems. Consequently, there is significant potential for further development and application of marine algal toxins, warranting more in-depth research in this area in the future.

This section explores the potential pharmacological applications of marine algal toxins and their mechanisms of action. Despite the significant toxicity of these toxins, recent studies have shown that they have important pharmacological activities in anti-tumor, analgesic, neuroprotective, and immune regulatory applications. For example, paralytic shellfish toxin (PSP) and Ocadaic acid (OA) have demonstrated inhibitory effects on certain cancer cells, while tetrodotoxin (TTX) is an important candidate for the treatment of neuropathic pain, due to its blocking effect on sodium ion channels. In addition, certain marine algal toxins, such as algal polysaccharide and Yao mycotoxin (YTX), show neuroprotective and immunomodulatory potential for the treatment of neurodegenerative diseases and allergic reactions. However, although these toxins show a wide range of pharmacological applications, their safety needs to be further explored. Therefore, future research should aim to optimize the therapeutic formulations and application strategies for these toxins in order to reduce the risk of toxicity and ensure their safety and efficacy in clinical use. Figure 2 shows the pharmacological activity and mechanisms of marine algal toxins.

## 5. Discussion and Conclusions

Marine algal toxins, as a class of highly bioactive natural products, are usually considered harmful compounds, and their contamination in marine-derived products should be avoided. However, emerging studies indicate that marine algal toxins have broad application prospects in the fields of medicine and biological sciences. This review summarizes the classification, toxicological and pharmacological activities, and potential applications of marine algal toxins, emphasizing their importance in the fields of medicine and health sciences. Indeed, marine algal toxins have shown promising potential in treating major diseases such as cancer, as well as in providing benefits like pain relief, neuroprotection, and immunomodulation. However, developing marine algal toxins into drugs or health-promoting products, and discovering medicinal applications for them, still requires a significant amount of further research, and we face many challenges.

Firstly, the chemical structures of marine algal toxins are very complicated. It is tough to produce, extract, purify, identify, and assay marine algal toxins. Indeed, different marine algal toxins require different processing methods. Future research should focus on developing efficient and safe production, extraction, and purification techniques. Synthetic biology may be useful for producing marine algal toxins, however, the processes of extracting and purifying these marine algal toxins remain a huge challenge. Concentration assays for marine algal toxins are also very important, since contamination in marine-derived products or the marine environment should be avoided. The identification and assay of marine algal toxins are also key for quality control in pharmaceutical development. The latest detection methods for marine algal toxins are rapidly evolving, driven by advancements in analytical chemistry and molecular biology. Techniques such as liquid chromatography–mass spectrometry (LC-MS), enzyme-linked immunosorbent assays (ELISAs), and molecular biosensors are being refined to increase sensitivity, specificity, and throughput. These methods are crucial for accurately monitoring toxin levels in marine environments and ensuring the safety of marine-derived products. Continuous improvement and validation of these detection technologies are essential to keep pace with the dynamic nature of algal blooms and toxin production; more sensitive, specific, rapid, and high-throughput methods will be required in the future.

Secondly, there are safety concerns linked to the fact that marine algal toxins have wide pharmacological activities. Marine algal toxins may affect various biological receptors and metabolic processes when they enter organisms due to their unique chemical structures, which may lead to a wide range of functional groups and toxicological and pharmacological properties. Their chemical structures allow them to bind specifically to biomolecules, thus regulating a variety of physiological processes, which may lead not only to toxicological effects, such as neurotoxicity and cytotoxicity, but also to beneficial pharmacological effects, such as anti-tumor and anti-inflammatory effects. Therefore, studying the dual effects of these toxins is important to fully understand their applications to medicine [131]. Thus far, studies have shown that marine algal toxins have potential applications for their anti-tumor, analgesic, neuroprotective, and immunomodulatory activities. Some algal toxins have been found to induce apoptosis in cancer cells, reduce pain, protect neurons from damage, and regulate immune responses. However, the high pharmacological potency of these toxins is often accompanied by safety concerns. Researchers are exploring the interactions between the pharmacological activities and toxicity of these marine algal toxins in depth in order to ensure that side effects are minimized while their pharmacological benefits are exploited. Balancing the beneficial effects of these toxins with their potential risks is key to current research. Sometimes, we can use topical instead of systematic applications of marine algal toxins in vivo, through which some side effects could be better controlled. Specifically, dosages are very important. If the dosages required to achieve pharmacological effects are much lower than those that cause toxicological effects, the toxins would be considered safe for clinical applications, however, investigations into how to use marine algal toxins, including determination of the dosages of pharmacological and toxicological effects in vivo, are still lacking, and further data regarding the balance of safety and efficacy are still required.

Thirdly, investigations into the pharmacological activities and mechanisms of the toxins discussed herein are still in their infancy. As described above, although marine algal toxins show some significant toxicological and pharmacological effects, any further pharmacological functions require additional investigation. A large-scale and systematic evaluation of their pharmacological activities should be conducted, either in vivo or in vitro, whereby additional potential clinical applications may be found. Although we have found that some algal toxins can exert effects by targeting certain protein molecules, the molecular mechanisms of most of the investigated toxins remained unclear, thus, in-depth studies of algal toxin interactions with their targets are important directions for future research. Bioinformatics or network pharmacology may be useful directions for finding potential molecular targets for our reference, however, wet experiments should be conducted in order to validate those potential targets, and multi-omics methods (determining the levels of proteins, mRNAs, metabolism, or single cells) should be utilized to determine their mechanisms of action. Future research should continue to explore these areas to better understand and apply these natural products from marine sources.

Lastly, the potential development of marine algal toxins into pharmaceutical products is a huge and systematic engineering effort, and necessary interdisciplinary collaboration is required. By integrating knowledge and technologies from chemical engineering, material sciences, synthetic biology, bioinformatics, pharmacology, toxicology, medicine, and the marine and environmental sciences, there are prospects for the development of new algal toxin drugs or therapeutic methods. Regular idea exchanges and collaboration among researchers who have different research backgrounds to establish clear guidelines and conduct comprehensive basic and clinical trials will be critical to overcoming these challenges and ensuring the safe applications of these promising compounds in medicine.

In conclusion, marine algal toxins with diverse chemical structures have significant and wide pharmacological activities and show promising potential medicinal applications as health-promoting products for the treatment of many diseases, although they also have some toxicological activities or untoward responses that must be avoided or limited by using appropriate dosages or methods. The discoveries of the pharmacological activities of marine algal toxins and the development of potential applications in health enhancement largely promote the comprehensive utilization and development of marine resources, however, because of interdisciplinary requirements, we still face numerous technical challenges that need to be addressed systematically in the future.

## Figures and Tables

**Figure 1 ijms-25-09194-f001:**
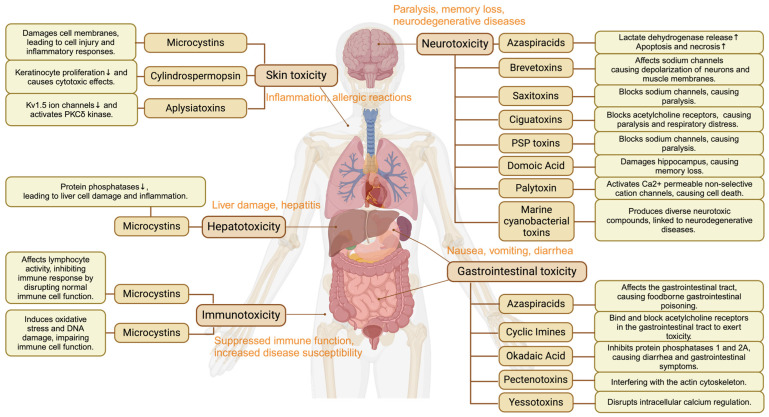
Toxicity and mechanisms of marine algal toxins.

**Figure 2 ijms-25-09194-f002:**
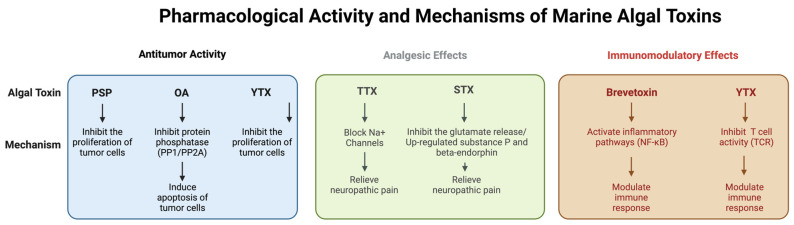
Pharmacological activities and mechanisms of marine algal toxins.

**Table 1 ijms-25-09194-t001:** Chemical classifications and sources of marine algal toxins.

Marine Algal Toxin Family	Main Components	Source	Chemical Structure *	Chemical Characteristics	Main Toxins	Toxic Mechanisms	LD50 in Mice (µg/kg)	References
Azaspiracids (AZAs)	Polyether toxins	Genus Azadinium	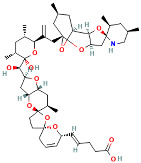 AZA-1	Molecular weight ~715 Da, 40 carbon atoms, 20 stereoisomeric centers, 9 ring structures, carboxyl group at the end, acidic	AZA-1, AZA-2, AZA-3, AZA-4, AZA-5, Aplysiatoxins (ATXs)	Alters calcium ion flow, causing abnormal intracellular calcium ion levels, leading to apoptosis and necrosis.	200	[28,29,30,31]
Brevetoxins (BTXs)	Polyether and cyclic aldehyde	Karenia brevis	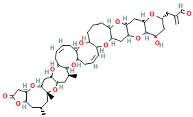 BTX-1	Molecular weight 800–900 Da, contains multiple conjugated double bonds and epoxy groups	BTX-1, BTX-2, BTX-3, BTX-6, BTX-9, Ciguatoxins (CTXs)	Acts on voltage-gated sodium channels, causing neuronal overexcitation	500	[34,36]
Cyclic Imines (CIs)	Cyclic amine groups combined with amino acids or peptides	Genus Alexandrium	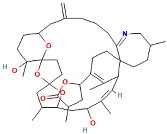 Spirolide B	Molecular weight 300–500 Da, contains a cyclic amine group and multiple cyclic ethers	Spirolide B, Gymnodimines, Pinnatoxins, Spirolides, Gymnodimines, Pteriatoxins, Palytoxin (PLTX)	Blocks acetylcholine receptors, affecting neural signal transmission	-	[37]
Domoic Acids (DAs)	Similar to glutamic acid	Genus Pseudo-nitzschia	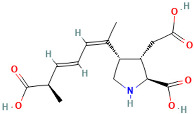 Domoic Acid	Molecular weight 311.1 Da, contains a cyclohexane ring and multiple carboxyl groups	Domoic Acid, Allomeric Acid, Isodomoic Acid, Prodomoic Acid	Mimics glutamic acid, causing neuronal overexcitation and damage	3600	[39,40]
Okadaic Acids (OAs)	Polyketide system	Genus Dinophysis	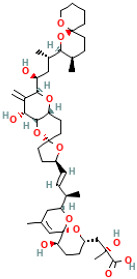 Okadaic Acid	Molecular weight 805.1 Da, contains multiple cyclic ester and carboxyl groups	Okadaic Acid, Dinophysistoxin-1 (DTX-1), Dinophysistoxin-2 (DTX-2), Dinophysistoxin-3 (DTX-3), Microcystin, Cylindrospermopsin (CYN)	Inhibits protein phosphatases, causing cytotoxicity	4000	[41,42]
Pectenotoxins (PTXs)	Polyether and ester bonds	Genus Dinophysis	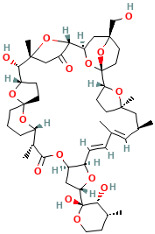 PTX-1	Molecular weight 800–900 Da, contains multiple cyclic ester and lactone groups	PTX-1, PTX-2, PTX-3, PTX-4, PTX-11	Affects the integrity of the cytoskeleton and cell membrane	-	[43]
Saxitoxins (STXs)	Multiple guanidine groups	Dinoflagellates (e.g., Gymnodinium, Alexandrium)	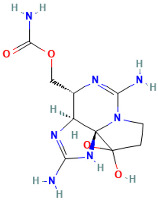 Saxitoxin	Molecular formula C10H17N7O4, molecular weight 299, contains multiple guanidine groups, potent neurotoxin	Saxitoxin (STX), Neosaxitoxin (NSTX), Decarbamoylsaxitoxin (dcSTX), Gonyautoxins (GTXs), C-11 Hydroxy Derivative, Cylindrospermopsin, Tetrodotoxin (TTX)	Potent neurotoxin	10	[44,45,46]
Yessotoxins (YTXs)	Multiple cyclic ethers and ester bonds	Protoceratium reticulatum	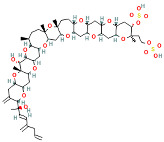 Yessotoxin	Molecular weight 1141.5 Da, contains multiple cyclic ethers and ester bonds	Yessotoxin (YTX), 45-Hydroxyyessotoxin, 41-Hydroxyyessotoxin, Yessotoxin Derivatives	Interferes with intracellular calcium ion balance and the cytoskeleton	500–750	[47,48,49]

* Chemical structure images sourced from PubChem (https://pubchem.ncbi.nlm.nih.gov/, accessed on 15 July 2024).

**Table 2 ijms-25-09194-t002:** Summary of marine algal toxins, their sources, and primary neurotoxic effects.

Toxins	Sources	Primary Toxic Effects and Mechanisms	References
Azaspiracids (AZAs)	Seafood contaminated by marine algae	Severe neurotoxicity; increases lactate dehydrogenase release, induces nuclear condensation, stimulates caspase-3 activity, and causes apoptosis and necrosis	[58,59,60]
Brevetoxins	Dinoflagellate Karenia brevis	High affinity for voltage-sensitive sodium (Na^+^) channels; causes ataxia and neurotoxic shellfish poisoning (NSP)	[34,61,62]
Cyclic imines	Various marine algae	Blocks acetylcholine receptors (nicotinic and muscarinic), disrupts neuromuscular transmission, causes paralysis and respiratory difficulty	[63]
Paralytic shellfish toxins (PSPs)	Prokaryotic freshwater cyanobacteria and marine dinoflagellates	Blocks voltage-gated sodium channels, causes paralysis	[64,65]
Saxitoxin	Freshwater cyanobacteria and marine dinoflagellates	Blocks voltage-gated sodium channels, prevents normal cell function, and causes paralysis	[7,10,66,67]
Tetrodotoxin (TTX)	Certain bacteria, cyanobacteria, and red algae	Binds to voltage-gated sodium channels, prevents neural transmission, and causes paralysis	[68,69]
Domoic acid	Diatoms of the genus Pseudo-nitzschia and red algae of the genus Chondria	Causes amnesic shellfish poisoning (ASP), damages memory regions like the hippocampus, decreases mitochondrial function, and increases oxidative stress	[71,73]
Ciguatoxins	Benthic microalgae	High affinity for voltage-gated Na^+^ channels; causes limb tingling, cold allodynia, and cardiovascular and respiratory failure	[5,76]
Marine cyanobacteria	Various marine cyanobacteria	Produces structurally diverse toxins; associated with neurodegenerative diseases such as Alzheimer’s and Parkinson’s; causes mild to severe health effects	[78]
Palytoxin (PLTX)	Marine organisms, particularly zoanthids	Activates Ca^2+^-permeable non-selective cation channels, increases intracellular Ca^2+^ concentration, and causes cell necrosis	[79,80,81]

**Table 3 ijms-25-09194-t003:** Pharmacological activities and mechanisms of marine algal toxins against cancer cells.

Algal Toxin	Cancer Cells	Dosage	Mechanism	References
Paralytic Shellfish Toxin (PSP)	Canine Hemangiosarcoma, PBMC	High dose, 100 μg/mL	Delays abdominal metastasis progression, increases CD14(+)/CD16(−) monocyte count	[18,19]
Okadaic Acid (OA)	Caco-2	48.8 nM	Inhibits PP1 and PP2A, induces tumor cell apoptosis	[20]
Okadaic Acid (OA)	Neuro-2a, NG108-15, MCF-7	11.20 nM, 13.34 nM, 46.95 nM	Inhibits PP1 and PP2A, induces tumor cell apoptosis	[119]
Dinophysistoxin-1 (DTX-1)	Neuro-2a, NG108-15, MCF-7	5.46 nM, 5.66 nM, 12.27 nM	Inhibits PP1 and PP2A, induces tumor cell apoptosis	[119]
Dinophysistoxin-2 (DTX-2)	Neuro-2a, NG108-15, MCF-7	Half of OA, 64.71 nM	Inhibits PP1 and PP2A, induces tumor cell apoptosis	[119]
Yessotoxin (YTX)	Melanoma, Lung Cancer, Colon Cancer, Acute Leukemia cells	High dose 100 μg/Kg	Induces apoptosis, shows sensitivity in some tumor cell lines, observed tumor tissue improvement in B16F10 model	[21,22]

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
