# Peer review of "Toxicological and Pharmacological Activities, and Potential Medical Applications, of Marine Algal Toxins"

_ijms, 2024, doi:10.3390/ijms25179194_

Round 1

Reviewer 1 Report

Comments and Suggestions for Authors

 Bioactive and application of Marine alga Toxins 

In general, article is not well organized.

-              The authors should improve and rewrite the abstract.

-              The manuscript does not present the problem statement, motivation and research objective (Review) erc.

The introduction section is not clear. This section should incorporate the research problem and critical discussions on the advantages and limitations of the proposed methods to solve this research problem. In addition, this section should add the main advantages and drawbacks of the proposed method in comparison with others reported in the literature.

Graphical abstract: low quality image (low resolution). 

The manuscript needs a lot of work. It should clearly detail each part of the process and show the complete study, additional comparative tables, and the design of experiments, which should be detailed and discussed

The discussion in each section and during the development of the manuscript is not well detailed, it is not discussed, and it does not clearly present the elaboration and comparison of the same. It lacks more scientific and clear support. however, it does not reflect, describe, or compare with respect to what; it does not justify or validate a scientific argument.

-              The authors need to state clearly the present problem (motivation of their work). 

-              The sections proposed in their review are crucial, and their relevance and timeliness need to be enhanced. It's also important to keep the content updated. 

-              Improve the conclusion section. 

-              the sections are incorrectly selected

-              the document should be improved, add more illustration as it is a review.

-              What are the main limitations or challenges of the proposed method?

-              The conclusions must be improved.

The authors should add figures and schematics

Conclusion

-              Improve the discussion section, compare it with other work, scope, and limitations, and reflect on its projection of the importance of the research achieved. 

Add more bibliographical references and do a more exhaustive reading of information. 

The manuscript must present a bibliographic reference between lines 46-48.

Authors should add in section 2 the chemical structure of each toxin family and then describe them in detail. 

missing references line 83 and in most of the manuscript 

Comments on the Quality of English Language

need to improve the writing 

improve their summary 

do not use short paragraphs 

correctly describe the information, advantages, disadvantages, limitations, 

be conclusive in your writing and information.

add good quality images 

improve the proposed sections 

make comparative tables 

Author Response

Dear Reviewer,

Thank you for giving us an opportunity to revise our manuscript, we appreciate editor and reviewers very much for their positive and constructive comments and suggestions on our manuscript entitled “Bioactivity and Applications of Marine Algal Toxins” (ID: ijms-3146778).

According to the reviewers’ comments, we have made necessary modifications to our manuscript and added additional explanations to make our results convincing. The revised paragraphs (sentences) are labeled in different colors.

The revised paragraphs based on REVIEWER#1 is labeled in yellow.

We would like to express our great appreciation to you and reviewers for comments on our paper. Looking forward to hearing from you.

Thank you and best regards.

Sincerely,

Xinyu Gao

Weidong Xie

Tel: +86-755-26036086

Reviewer 2 Report

Comments and Suggestions for Authors

Title and Abstract

Line 2-3: Specify the methodologies or analytical techniques reviewed in the paper to provide clarity about the scope.

Line 21-23: Ensure that the keywords cover all major topics discussed in the paper. Consider adding terms like "pharmacological mechanisms" or "environmental impact."

Introduction

Line 24-25: Expand the introduction to include a brief history of marine algal toxins to provide context.

Line 28: Define any acronyms at their first occurrence, such as WHO, IOC-UNESCO.

line 33: toxic molecules that pose high toxicity-->rephrase, 2 x toxic

line 36 and line 37: impact on human health: 2 x same info!

Chemical Classification and Sources of Marine Algal Toxins

Line 73-74: Consider adding visual aids like diagrams to represent the chemical structures of the toxins discussed.

Line 85: Include a brief comparison of the toxicity levels between different marine algal toxins for a more comprehensive overview.

Toxicity and Mechanisms of Marine Algal Toxins

Line 198-199: Expand on the discussion of neurotoxicity by adding more recent studies that illustrate the mechanisms of action.

Line 205-207: Include a summary table for better readability that lists each toxin, its source, and its primary toxic effects.

Pharmacological Activity and Mechanisms

Line 451-454: Add references to recent studies that support the health benefits mentioned, particularly for emerging research areas.

Line 458-461: Discuss potential adverse effects of using these toxins in therapeutic settings to provide a balanced view.

Discussion and Conclusion

Line 485: Rephrase the sentence to better reflect both the potential benefits and the risks associated with marine algal toxins.

Figures and Tables

Line 83-84: Include tables summarizing key information, such as toxicity levels, pharmacological effects, and chemical structures of the toxins.

Potential Additions

Line 485: Add a brief discussion on the latest detection methods for marine algal toxins, as this is a rapidly evolving area.

Line 489: Consider discussing the regulatory challenges associated with the therapeutic use of marine algal toxins.

Comments on the Quality of English Language

an interesting paper, in adequate English, only minor improvements needed for the language

Author Response

Dear Reviewer,

Thank you for giving us an opportunity to revise our manuscript, we appreciate editor and reviewers very much for their positive and constructive comments and suggestions on our manuscript entitled “Bioactivity and Applications of Marine Algal Toxins” (ID: ijms-3146778).

According to the reviewers’ comments, we have made necessary modifications to our manuscript and added additional explanations to make our results convincing. The revised paragraphs (sentences) are labeled in different colors.

The revised paragraphs based on REVIEWER#2 is labeled in yellow.

We would like to express our great appreciation to you and reviewers for comments on our paper. Looking forward to hearing from you.

Thank you and best regards.

Sincerely,

Xinyu Gao

Weidong Xie

Tel: +86-755-26036086

Reviewer 3 Report

Comments and Suggestions for Authors

This paper reviews the bioactivity of marine algal toxins. This is a well structured and complete review about this subject. Some minor optimizations of this reviews are suggested:

- Previous reviews within this subject should be cited and the novelty of the present paper highlighted. 

- The time interval of the papers under review should be defined in the last paragraph of the introduction.

Author Response

Dear Reviewer,

Thank you for giving us an opportunity to revise our manuscript, we appreciate editor and reviewers very much for their positive and constructive comments and suggestions on our manuscript entitled “Bioactivity and Applications of Marine Algal Toxins” (ID: ijms-3146778).

According to the reviewers’ comments, we have made necessary modifications to our manuscript and added additional explanations to make our results convincing. The revised paragraphs (sentences) are labeled in different colors.

The revised paragraphs based on REVIEWER#3 is labeled in yellow.

We would like to express our great appreciation to you and reviewers for comments on our paper. Looking forward to hearing from you.

Thank you and best regards.

Sincerely,

Xinyu Gao

Weidong Xie

Tel: +86-755-26036086

Round 2

Reviewer 1 Report

Comments and Suggestions for Authors

 In general, the manuscript not is well organize, don’t have format , and style of the journal . This manuscript must be significantly

1.- The introduction section must incorporate the research problem to resolve. In addition, this section must add the advantages and limitations of the works proposed in the literature to resolve the research problem. 

2.- The introduction section must consider the innovation or scientific contribution and advantages of the proposed in comparison with others reported in the literature

3.- Authors are recommended to add a graphical abstract. 

4.- Authors are recommended to keep order when citing in their manuscript and to add more references (to make a more solid background).ans style of journal

5,. the proposed sections in the manuscript could be more relevant and better organized. In the  review

6.-  the authors should add more figures and diagrams. 

7.- The authors must add more discussions in all sections

8.- The authors must incorporate the challenges of the proposed in the conclusion

9.-  What are the future research works?

Comments on the Quality of English Language

The authors must improve the English grammar and style of all the sections of the manuscript

check the style and grammar of the journal 

as well as how to cite 

check the guide for authors 

Author Response

(The authors gave the same response as above.)
